# Antimicrobial Stewardship in the Hospital Setting: A Narrative Review

**DOI:** 10.3390/antibiotics12101557

**Published:** 2023-10-21

**Authors:** Helen Giamarellou, Lamprini Galani, Theodoros Karavasilis, Konstantinos Ioannidis, Ilias Karaiskos

**Affiliations:** 11st Department of Internal Medicine-Infectious Diseases, Hygeia General Hospital, 4 Erythrou Stavrou & Kifisias, Marousi, 15123 Athens, Greece; l.galani@hygeia.gr (L.G.); tkaravasilis@hygeia.gr (T.K.); ikaraiskos@hygeia.gr (I.K.); 2Clinical Pharmacists, Hygeia General Hospital, 4 Erythrou Stavrou & Kifisias, Marousi, 15123 Athens, Greece; kioannidis@hygeia.gr

**Keywords:** stewardship, antibiotics, resistance, preauthorization, restriction, handshake, audit, feedback, de-escalation, surgical prophylaxis

## Abstract

The increasing global threat of antibiotic resistance, which has resulted in countless fatalities due to untreatable infections, underscores the urgent need for a strategic action plan. The acknowledgment that humanity is perilously approaching the “End of the Miracle Drugs” due to the unjustifiable overuse and misuse of antibiotics has prompted a critical reassessment of their usage. In response, numerous relevant medical societies have initiated a concerted effort to combat resistance by implementing antibiotic stewardship programs within healthcare institutions, grounded in evidence-based guidelines and designed to guide antibiotic utilization. Crucial to this initiative is the establishment of multidisciplinary teams within each hospital, led by a dedicated Infectious Diseases physician. This team includes clinical pharmacists, clinical microbiologists, hospital epidemiologists, infection control experts, and specialized nurses who receive intensive training in the field. These teams have evidence-supported strategies aiming to mitigate resistance, such as conducting prospective audits and providing feedback, including the innovative ‘Handshake Stewardship’ approach, implementing formulary restrictions and preauthorization protocols, disseminating educational materials, promoting antibiotic de-escalation practices, employing rapid diagnostic techniques, and enhancing infection prevention and control measures. While initial outcomes have demonstrated success in reducing resistance rates, ongoing research is imperative to explore novel stewardship interventions.

## 1. Introduction

The discovery of antibiotics and vaccines has been acknowledged worldwide as two of the most prominent developments in Clinical Medicine. The cascade of antibacterial agents started in the 1940s, when sulfonamides, penicillin, and streptomycin were introduced in the market. There is no doubt that antibiotics have radically changed the future of medicine by curing once lethal infections. Nevertheless, it was established early on that a bacterium, when exposed to antibiotics, develops survival strategies. This raised an unprecedented concern in the history of antibiotic usage for the imperative need for judicious prescription to preserve their unparalleled value and efficacy [1,2,3,4,5,6,7].

It was Sir Alexander Fleming who made the following cautionary statements in a *New York Times* article, on 26 June 1945: “… the microbes are educated to resist penicillin and a host of penicillin-fast organisms is bred out… In such cases the thoughtless person playing with penicillin is morally responsible for the death of the man who finally succumbs to infection with the penicillin-resistant organism. I hope this evil can be averted” [1]. Fleming, in light of his statement, emerged as a discerning voice to the medical community. While he expressed concerns that bacteria could develop resistance to penicillin due to irresponsible use, he also maintained a hopeful perspective, believing that responsible and prudent use of antimicrobial agents by physicians could help prevent such detrimental consequences. It was only in 1945 that penicillin resistance was recognized in *Staphylococcus aureus* [2], whereas methicillin-resistant *Staphylococcus aureus* (MRSA) strains emerged in 1961 [3] to be followed more than 30 years ahead by penicillin-resistant *Streptococcus pneumoniae* and *Neisseria gonorrhea* [4,5], as well as vancomycin-resistant *Enterococcus faecium* (VRE) and vancomycin-resistant *S. aureus* (VRSA) [6,7].

## 2. Redefining the Problem of Antimicrobial Resistance in the 21st Century

The rise of the 21st century coincided with the emergence of multidrug-resistant (MDR) Gram-negative bacteria, including extended spectrum beta-lactamases (ESBL) producing Enterobacterales, extensively drug-resistant (XDR) carbapenemase-producing *Klebsiella pneumoniae* and other Enterobacterales, as well as *Pseudomonas aeruginosa* and unfortunately pan-drug-resistant (PDR) *Acinetobacter baumannii* [8,9,10,11,12,13]. The current classification of antibiotic-resistant bacteria is nowadays categorized into three classes [13]: (1) MDR indicating non-susceptibility to at least one antibiotic within three or more antibiotic categories; (2) XDR indicating non-susceptibility to at least one antibiotic in all but two or fewer antibiotic categories; and (3) PDR indicating non-susceptibility to all antibiotics in all antibiotic categories. The antibiotic categories’ classes are penicillins, cephalosporins, monobactams, carbapenems, aminoglycosides, fluoroquinolones, folate pathway inhibitors, glycylcyclines, phosphonic acids, polymyxins, phenicols, and tetracyclines. Recently, a new definition of resistance for Gram-negative infections defined as difficult-to-treat resistance (DTR) has been proposed, indicating treatment-limiting resistance to all first-line antibiotics, including all β-lactams and fluoroquinolones [11,14]. Cassini et al. [15] reported that infections attributed to antibiotic-resistant bacteria in the European Union (EU) and the European Economic Area for the year 2015 reached a staggering 671,689 cases, resulting in 33,110 attributable deaths. Italy and Greece emerged as the nations within the EU with the highest mortality, with a median number of cases reaching 201,584 and 18,472, respectively. The median number of deaths in Italy stood at 10,762, while in Greece, it was 1626. [15]. The burden of global resistance in 2019 has been estimated as 4.95 million deaths of which 1.27 million were directly attributable to bacterial antimicrobial resistance [16]. Within the United States, the CDC has estimated that 2,868,700 people were infected with antibiotic-resistant bacteria and fungi ending with 35,900 deaths each year [17]. 

## 3. Facing the End of Antibiotics

Facing the prophetic announcement made by Jim O’Neill in 2014, where he foresaw an annual global death toll of 10 million people by 2050 due to infections caused from antibiotic-resistant bacteria, involves antimicrobial resistance becoming the leading cause of mortality. This scenario was also accompanied by a staggering economic burden, reaching up to an astonishing USD 100 trillion. The question that arises is how can this alarming situation be addressed and averted? [18]. Even in June 2023, WHO Western Pacific announced that antimicrobial resistance is expected to cause 5.2 million deaths in the Western Pacific by 2030 [19]. Otto Cars, a Swedish pioneer in the field of antimicrobial resistance, has been continuously engaged in the fight against antimicrobial resistance, a battle he has labeled as “An Ethical Challenge”. As far back as 2015, he issued a warning, noting that “the limited effectiveness of antibiotics leads to questions about our responsibility for the well-being of further generations”. In essence, he highlighted that “Ethics deals with what we ought to do or ought not to do” [20]. Therefore, the crucial question arises: what actions should or should not be taken in the ongoing struggle against antimicrobial resistance? 

Evidence demonstrates that at least in hospital settings worldwide, the application of hospital-based programs dedicated to the rational use of antibiotics is urgently required, referred to as Antimicrobial Stewardship Programs (ASPs) [21]. A narrative review of relevant studies was conducted using the PubMed/MEDLINE, Scopus, and Web of Science databases (from 1970 up to June 2023). The keywords used alone or in combination were antibiotic stewardship, Antimicrobial Stewardship, antimicrobial resistance, Infectious Diseases, antibiotic prescribing, infection control, healthcare-associated infections, antibiotic policy, antimicrobial intervention, antibiotic consumption, restrictive antibiotic formulary, preauthorization, prospective audit and feedback, post-prescription review, de-escalation, surgical prophylaxis, surgery, diagnostic stewardship, rapid diagnostic techniques, nurse and stewardship, ICU patients, and critically ill patients. Information regarding Antimicrobial Stewardship innovations was included. Studies that lacked an ASP intervention, did not assess primary outcomes, or were conducted in animal settings were excluded. Full text and abstract screening as well as review articles were searched. A total of 550 studies were generated by searching. Out of those, 102 articles were sought for retrieval. 

In this comprehensive narrative review, the current state of evidence was thoroughly investigated concerning antibiotic stewardship within hospital settings. Our analysis not only explored the global landscape but also placed a specific focus on practical applications and real-world examples of Antimicrobial Stewardship practices in Greece. The term “Antimicrobial” should be used interchangeably with “Antibiotic” in this context, as our focus is solely on the stewardship of antibiotics. 

## 4. Antimicrobial Stewardship in the Hospital Setting: Definitions—Personnel

The current definition of Antimicrobial Stewardship (AS) refers to ‘‘Systemic measurement and coordinated interventions designed to promote the optimal use of antimicrobial agents including their appropriate selection, dosing, route and duration of administration for all patients including those who are critically ill or immunocompromised’’ [21,22,23,24]. The definition applies not only to antibiotics but also to antifungals and antivirals [21], which are out of the scope of this review. Therefore, the primary goal of AS is the optimization of the clinical outcome while minimizing unintended consequences of antibiotics, including toxicity, the selection of pathogenic bacteria (such as *Clostridioides difficile*), and the emergence of resistance as well as the reduction in the existing resistance rates [21,22]. The multifaceted purposes of AS have been defined in a consensus statement from the Infectious Diseases Society of America (IDSA), the Society for Health Epidemiology of America (SHEA), and the Pediatric Infectious Diseases Society (PIDS) [22,23].

To establish and operate an effective AS program, the organization of a multidisciplinary team is required. According to IDSA and SHEA guidelines of 2007, an engaged Infectious Diseases (ID) physician and a clinical pharmacist with Infectious Diseases training, as well as a clinical microbiologist, an Information System Specialist, an Infection Control Professional, and a hospital epidemiologist, should be included as core members [25]. Since AS is a medical staff function, as well as an important component of patients’ safety, it should be directed by an ID physician or by the latter specialist plus a clinical pharmacist with Infectious Diseases training [26]. However, depending on the hospital’s size, smaller teams can be formed, with the essential presence of the ID physician and the clinical pharmacist. These core team members can be further complemented with the inclusion of a surgeon and a skilled nurse [21,26]. The ASP leader should possess strong leadership skills, to motivate, inspire, and influence others and not fear confrontation. 

A comprehensive evidence-based AS program aiming to combat antimicrobial resistance should include the following important issues [21,26]: The collaboration of the AS Team members with the Hospital Infection Control Committee, the pharmacy, and all other hospital therapeutic committees with emphasis on official surgery and Hematology/Oncology representation, whenever required.The support and collaboration of hospital administrators with the medical staff leaders.The negotiation of the ID physician and the head of the pharmacy with the hospital administration, regarding authority and compensation in relation to the expected outcome of the AS program.The hospital administrative support to measure antibiotic consumption and to track use on an ongoing basis through the necessary infrastructure.

In 2014, the CDC recommended that all acute care hospitals in the United States should implement AS programs. Therefore, the so-called “Core Elements of Hospital Antibiotic Stewardship Programs” by the CDC to support implementation of AS, updated in 2019, was released, discussing lessons learned from a 5-year (2014–2018) accumulated experience [27]. In addition, AS Core Elements for small and critical hospitals were also evaluated and reported [28]. In summary, the updated Core Elements concern the following:Hospital Leadership Commitment referring to “Dedicated necessary, human, financial and information technology resources” with emphasis on “the necessity of AS programs leaders having dedicated time and resources to operate the program effectively”. Additionally, support from the senior leadership of the hospital, particularly the Chief Medical Officer, the Chief Nursing Officer, and the Director of Pharmacy, is considered critical for the success of AS programs. To be also pointed out is that the hospital leaderships play a critical role in obtaining the resources required to accomplish AS targets.Accountability regarding the appointment of a leader or co-leaders, i.e., a physician and a pharmacist trained in Infectious Diseases, both responsible for the program application and the outcome.Drug Expertise appointment, aiming to lead implementation effort to improve antibiotic use.Tracking antibiotic prescribing, impact of interventions, side effects, and resistance patterns, elements vital for continuous assessment.Reporting at regular time intervals information on antibiotic consumption and incidence of resistance rated to prescribers, pharmacists, nurses, and hospital leadership as well.Education targeting prescribers, pharmacists, and nurses regarding antibiotic resistance, optimal prescribing, and adverse reaction of antibiotics.Action requiring implementation of interventions focusing on a restrictive antibiotic formulary and preauthorization as well as in a prospective audit and feedback, which are both characterized as “priority interventions”.

## 5. Antibiotic Stewardship in the Hospital Setting: Strategies of Implementation 

When initiating an AS program, the first action should be the calculation of the specific needs of the involved personnel as well as the strategies to be applied [21,29]. Firstly, it is imperative to identify the key personnel who will be actively engaged in the program. Following this, a thorough evaluation of the existing knowledge and training levels of these personnel is crucial. Resource allocation is another pivotal aspect, encompassing both human and material resources. This includes calculating the necessary staffing levels required for the program’s effective implementation. Data collection and analyses play a vital role and should encompass information pertaining to antibiotic usage, resistance patterns, and clinical outcomes. Additionally, it is essential to assess how the AS program will integrate with pre-existing healthcare systems and processes, ensuring seamless alignment. Effective communication and collaboration channels among the personnel involved must be established, alongside defining strategies for monitoring and ensuring compliance with AS guidelines and policies. Lastly, a comprehensive risk assessment should be conducted to identify potential challenges in implementing the AS program, with a clear delineation of the personnel’s role in mitigating these risks [21,29]. The AS Task Force of the United States Department of Veterans Affairs validated a staffing calculator in order to define the personnel needs for an efficient AS program. It was concluded that per 100 occupied beds, 1.0 full-time equivalent (FTE) pharmacist and 0.25 FTE physicians were required, whereas for hospitals with 301–500 beds, 3–5 pharmacists and 0.75–1.25 physicians, both FTE, were indispensable [30]. However, in another less rigorous cross-sectional survey, it was suggested that only 1.2 and 0.4 FTE pharmacists and physicians, respectively, would be required for an institution with 301 to 500 beds [31].

Three interventions have been prioritized by the CDC based on Antimicrobial Stewardship [21,22]:

### 5.1. Restrictive Antibiotic Formulary and Preauthorization

The restrictive antibiotic formulary and preauthorization are strategies employed as part of ASPs, which aim to ensure the appropriate use of antimicrobial agents in an era of increasing antimicrobial resistance. The restrictive antibiotic formulary strategy involves maintaining a list of approved antibiotics for use within a healthcare institution. The preauthorization strategy requires healthcare providers to obtain approval before prescribing certain antimicrobial agents [32]. 

In 2017, Goff et al. [33] proposed a terminology shift, advocating for the replacement of “restricted” with “protected” in the context of ASPs. This change aligns with the heightened responsibilities associated with ASPs, which should aim to safeguard not only newer antibiotics but also older ones, particularly when the latter are linked to “collateral damage,” as observed with fluoroquinolones [33].

Before the administration of “protected antibiotics,” the prerequisite is obtaining approval from an ID physician, or a pharmacist trained in Infectious Diseases. This program yields several advantages, including the reduction in unnecessary or inappropriate antibiotic usage, optimization of empiric selections, direct control over antibiotic utilization as a passive prescribing barrier, and cost reduction. It also serves as a platform for educating individual prescribers in appropriate antimicrobial chemotherapy, influencing subsequent antibiotic usage positively [21].

However, in cases involving patients with septic shock or those with risk factors suggestive of underlying XDR Gram-negative bacilli [13] as potential pathogens, the empiric administration of “Protected Antibiotics” is recommended while awaiting culture results, as illustrated in Table 1. 

Among the primary drawbacks of this approach is the potential barrier it poses to the autonomy of prescribers in clinical decision making [34]. Additionally, it may inadvertently lead to increased prescriptions of non-restricted antibiotics, a phenomenon known as the “Squeezing the Balloon” effect [35]. Furthermore, the preauthorization process can be time-consuming, and the verbal nature of the permission may sometimes result in incorrect recommendations, necessitating subsequent chart review [36]. Therefore, it is advisable to implement monitoring procedures for all prescribed antibiotics after the introduction of formulary restrictions [36]. It is essential that the ID physician responsible for granting approval possesses the requisite skills and qualifications, with real-time availability being a crucial factor in the process.

### 5.2. Prospective Audit and Feedback (PAF) (or Post-Prescription Review)

Highly trained personnel, often led by an ID physician, engage in the routine review of antimicrobial orders. The frequency of these reviews typically ranges from daily assessments to twice-weekly evaluations, a schedule determined with the hospital’s size. The purpose of these reviews is to furnish prescribers with written or verbal recommendations concerning their antibiotic selections, thus preserving the autonomy of the prescribing clinicians. It is worth noting that PAF has demonstrated its effectiveness in curbing the inappropriate use of antibiotics across various healthcare settings, encompassing ICUs, pediatric facilities, and community hospitals [37]. In community hospitals, PAF programs conducted thrice-weekly have proven to be successful in reducing antibiotic usage and generating cost savings [38,39]. However, due to the substantial time and labor commitment associated with PAF, some suggest that it could be targeted at specific patient groups. These include ICU patients, those receiving broad-spectrum, potentially with high toxicity, or multiple antibiotics, and cases requiring prolonged therapy [21]. An extensive meta-analysis involving more than 14,000 ICU patients across six observational studies revealed that despite the reduction in antibiotic usage with PAF, there was no observed difference in mortality rates [40]. The initial post-prescription review should ideally commence 2–3 days following the initiation of antibiotic therapy and encompass a comprehensive assessment, including examination of culture results and susceptibility reports, evaluation of C-reactive protein (CRP) and procalcitonin levels, scrutiny of the empiric therapy dosage and administration modalities, being grounded in pharmacokinetic–pharmacodynamic principles, determination of the appropriate duration of therapy, assessment of fecal colonization results for XDR bacteria in high-risk patients, consideration of de-escalation, escalation, or discontinuation of antibiotics, as well as measurement of the time interval between the administration of the initial antibiotic dose and the onset of septic shock [22]. Furthermore, PAF offers distinct advantages, including enhanced clinical guidance for prescribers, flexibility in timing, and educational benefits, all while maintaining prescribers’ decision-making autonomy [22]. Nonetheless, it should be acknowledged that participation in PAF is voluntary, the process is resource-intensive, and, importantly, prescribers often exhibit reluctance to alter antibiotic regimens once patients have shown a positive response to the initially administered therapy [22].

“Handshake Stewardship” introduces an innovative approach focused on engaging in ‘face-to-face’ interactions with each department’s staff individually. This approach has demonstrated its ability to enhance the impact of PAF while building collegial relationships [41,42]. Notably, “Handshake Stewardship” operates without the need for preauthorization. Under this approach, the ID physician and pharmacist undertake a comprehensive review of all antimicrobial prescriptions at two key junctures: 24- and 48–72-h post-prescription. Subsequently, they conduct daily rounds during which they personally meet with clinicians, employing a metaphorical ‘perfect handshake’ to initiate interactions before offering their recommendations and interventions. On average, one physician and one pharmacist dedicate approximately 1 h per day to reviewing around 50 antibiotic orders (25 each) before proceeding with “Handshake Stewardship” rounds. This rigorous process has led to a notable 23% reduction in antibiotic utilization [41,42]. It is important to acknowledge that for larger hospitals, the requirement of daily review of all antibiotics may pose logistical challenges. Nonetheless, modified versions of “Handshake Stewardship” have been proposed, particularly suitable for smaller hospitals. These adaptations may involve targeting specific units, selected antibiotics, or focusing on particular Infectious Diseases [28,37]. Additionally, larger hospitals may consider bolstering their ID physician resources [41,42]. The unique advantages of “Handshake Stewardship” lie in its symbolic gesture of bringing individuals together, even those with differing perspectives, by conveying trust and equality. This approach instills confidence and demonstrates a commitment to excellence. Nevertheless, it is crucial for the AS leader to remember that the first and most vital ‘handshake’ must occur with the hospital administration. 

Few studies have compared PAF and preauthorization with conflicting results. In one study, PAF was associated with greater reduction in antibiotics, whereas in another, transition of preauthorization to PAF transaction was linked to an increase in the utilization of broad-spectrum antibiotic prescription [43,44]. Despite the lack of appropriate studies, it is rational that both methods can be applied since preauthorization helps to optimize initiation of more appropriate antibiotics, whereas PAF can optimize subsequent therapeutic handling. 

### 5.3. International Evidence on the Reduction in Antibiotic Resistance

Is there evidence to suggest that antibiotic resistance could be reduced through the implementation of a PAF approach? Preauthorization studies have consistently demonstrated a reduction in antibiotic usage and a decrease in antibiotic resistance, particularly among Gram-negative bacteria. Pioneering work by White et al. in 1997 revealed that implementing preauthorization requests for restricted antibiotics in a county teaching hospital resulted in a remarkable 32% reduction in total parenteral antibiotic expenditures (*p* < 0.01). This reduction was accompanied by an increase in the susceptibility of Gram-negative bacilli without any discernible impact on hospital length of stay or mortality rates [45]. Notably, *Pseudomonas aeruginosa* susceptibility to imipenem increased from 65% to 83% (*p* ≤ 0.1) in ICU isolates. Another study by Rehal et al. [46] introduced preauthorization requirements for cephalosporin prescriptions. While this led to a reduction in ceftazidime-resistant *Klebsiella* cases, it also resulted in increased imipenem use and a subsequent 69% rise in *P. aeruginosa* resistance rates. This underscores the importance of carefully monitoring antibiotic resistance rates and dosing when implementing restrictive measures. Furthermore, it is essential to highlight potential communication challenges when obtaining approval for restricted antibiotics via telephone calls [47]. However, the development of a computerized approval system based on specific indications for restricted antibiotics demonstrated significant reductions in antibiotic consumption and an increase in Pseudomonas susceptibility over a 2-year period [48]. Reed et al. at the Ohio State University Wexner Medical Center implemented a well-organized computerized system, replacing imipenem with doripenem, which required prior authorization [32]. Despite the approval rate of 91% for doripenem requests, its use significantly decreased over a 10-month period (11 antimicrobial days/1000 patient days vs. 27 antimicrobial days/1000 patient days, *p* = 0.0008), without causing any ‘squeezing of the balloon’ effect of other antibiotics like ertapenem, cefepime, or piperacillin–tazobactam. In a retrospective multicenter study, Pakyz et al. evaluated the impact of carbapenem restriction over 5 years in eight university teaching hospitals on the susceptibility of *P. aeruginosa* isolates. Their findings showed that fewer carbapenem prescriptions (*p* = 0.04) corresponded to a lower incidence of carbapenem-resistant *P. aeruginosa* (*p* = 0.01) throughout the study period [49]. While restrictive antibiotic policies have shown promising results, some experts argue that, given the frequent empirical prescription of carbapenems in septic or high-risk patients, initial doses should be allowed while awaiting official approval to avoid potentially worsened patient outcomes [50].

PAF interventions have also demonstrated the ability to reduce antibiotic resistance and rates of *Clostridioides difficile* infection (CDI) without negatively impacting patient outcomes [51,52,53,54]. In a community hospital setting, PAF led to a 22% decrease in parenteral broad-spectrum antibiotics, subsequently reducing nosocomial infections attributed to antibiotic-resistant Enterobacterales over a 7-year period [51].

Comparing restriction with PAF as persuasive measures, a meta-analysis of 52 interrupted time series in a Cochrane review found that restrictive interventions had a statistically greater impact on reducing antibiotic-resistant bacteria within a 6-month timeframe (*p* = 0.001). This suggests that restrictions should be preferred when urgency is a factor [55]. In a comprehensive systematic review and meta-analysis covering 32 studies and 9,056,241 patient days from 1960 to 2016, the effects of ASPs on the incidence of infection and colonization with antibiotic-resistant bacteria and *C. difficile* infections in hospitalized patients were analyzed [56]. ASPs, including an antibiotic restriction, audit, or both, as well as the implementation of therapeutic guidelines, were primarily applied. ASPs were found to reduce the incidence of infections and colonization with multidrug-resistant Gram-negative bacteria (48%, *p* = 0.0428) and MRSA (37%, *p* = 0.0065), as well as *C. difficile* infections (32%, *p* = 0.0029). ASPs proved to be more effective when implemented alongside infection control measures (*p* = 0.0030), particularly hand hygiene interventions (*p* < 0.0002). The study acknowledged significant heterogeneity among the studies, partly attributed to the types of interventions and co-resistance patterns of the target bacteria. Nonetheless, these results provide valuable evidence for stakeholders and policy makers supporting the implementation of ASPs to alleviate the burden of infections caused by antibiotic-resistant bacteria.

### 5.4. The Greek Experience on Antimicrobial Stewardship

Greece is a European country possessing high resistance rates [11,15]. The restrictive antibiotic formulary and preauthorization are strategies employed as part of ASPs in many Greek hospitals. An illustrative example of a restricted antibiotic formulary is presented in Table 2. 

In a well-organized Greek study, which sought to assess the impact of a hospital-wide antibiotic restriction policy program on the resistance rates of nosocomial Gram-negative bacteria, spanning from 1998 to 2000, an innovative approach involving a restrictive formulary and preauthorization format was implemented at Sismanoglio Tertiary Hospital in the Athens area. This medium-sized facility achieved a remarkable 45% reduction in the consumption of restricted antibiotics, including aminoglycosides, third-generation cephalosporins, aztreonam, piperacillin/tazobactam, and ciprofloxacin, amounting to a reduction of 7.2 DDD per 100 patient days (*p* < 0.05). Notably, this reduction did not result in any discernible increase in the resistance rates of non-restricted compounds [57]. The noteworthy outcomes related to the reduction in resistance rates of *Pseudomonas aeruginosa* and *Klebsiella pneumoniae* are presented in Table 3.

In another Greek tertiary hospital, Zanion Hospital located at the Piraeus harbor, with a bed capacity of 427, the impact of a comprehensive ASP spanning 4 years from January 2016 to 2020 was evaluated. This program was overseen by a multidisciplinary team comprising two Infectious Disease physicians, two clinical microbiologists, two pharmacists, an intensivist, and two infection control nurses (all non-dedicated). The ASP incorporated a combination of a restrictive antibiotic formulary and prospective audit and feedback. An extensive data analysis encompassing the pre-intervention period of 2015 and the intervention period of 2016–2019 was conducted, covering aspects such as antibiotic consumption (expressed as DDD/100 bed days), rates of *C. difficile* infections, resistance rates, length of hospital stays, and annual costs. The results of this initiative were notable, with a significant average reduction of 8.23% in the consumption of restricted antibiotics (colistin, carbapenem, quinolones, tigecycline, glycopeptides, daptomycin, and linezolid) (*p* = 0.034). These reductions were followed by substantial decreases in resistance rates among *P. aeruginosa* and *K. pneumoniae* isolates, as elucidated in Table 4 and Table 5. Moreover, reductions in length of stay from 4.18 days in 2015 to 3.0 days in 2019, *C. difficile* infections per 1000 patients (1.47 in 2015 to 0.86 in 2019), and antibiotic costs (EUR 39.5 in 2015 to EUR 23.69 in 2019) were also observed [58].

In a study by Ntagiopoulos et al., the impact of an antibiotic restriction policy on antibiotic resistance of Gram-negative bacteria was examined in a Greek ICU [59]. The study focused on the epidemiology of infections caused by *A. baumannii*, *P. aeruginosa*, and *K. pneumoniae*, their resistance patterns, and antibiotic consumption. An antibiotic restriction policy was implemented, including restrictions on quinolones and ceftazidime. After an 18-month period, the same parameters were reassessed for an additional 6 months. The results showed a 92.5% reduction in the consumption of restricted antibiotics and a 55.4% reduction in overall antibiotic use. Susceptibility to ciprofloxacin significantly increased, but ceftazidime susceptibility increased only for P. aeruginosa. There were no differences observed in terms of infectious episodes, overall mortality, or ICU ecology.

### 5.5. Facility-Specific Guidelines

Based on the local epidemiological data and susceptibility patterns derived from reliable sources, it is imperative for AS programs to tailor their efforts to individual healthcare facilities. The primary objective should be the development of facility-specific therapeutic clinical practice guidelines that address the most prevalent infections. To facilitate the application of AS principles effectively, AS programs should initially incorporate the “4 Moments of Antibiotic Decision,” as elucidated by Tamma et al. [60]. These moments act as a practical framework for healthcare professionals:Moment 1: Does the patient have an infection that requires antibiotics?Moment 2: Have I ordered appropriate cultures before starting antibiotics? What empirical antibiotic therapy should I initiate?Moment 3: A day or more has passed. Can I stop antibiotics? Can I narrow therapy? Can I change from iv to per os therapy?Moment 4: What duration of antibiotic therapy is needed for the referred patient’s diagnosis?

These guidelines should offer healthcare facilities standardized recommendations for selecting the most appropriate antimicrobial agents, particularly for common infections such as community-acquired pneumonia (CAP), urinary tract infections (UTIs), skin and soft tissue infections (SSTIs), and a fever of unknown origin (FUO) in neutropenic patients, and the proper application of surgical prophylaxis. It is worth noting that a significant portion of antibiotics administered in United States hospitals are prescribed for CAP, UTIs, and SSTIs, presenting significant opportunities for enhancing prescription practices [22]. For CAP, interventions should prioritize improvements in diagnostic accuracy, tailoring therapy based on culture results, optimizing treatment duration, and ensuring adherence to guidelines. Multiple randomized controlled trials (RCTs) involving over 8400 CAP patients have demonstrated that shorter courses of therapy (3–5 days) are equally effective as longer durations (5–14 days) [21]. Moreover, vigilant monitoring of antibiotic durations is essential, with a general guideline indicating that in healthcare-associated pneumonia (HAP), including ventilator-associated pneumonia (VAP), antibiotic therapy should not exceed 7 days. However, for infections involving *Legionella* spp and *S. aureus*, a 2-week duration is recommended. Conversely, many patients receiving antibiotics are diagnosed with asymptomatic bacteriuria, which, except for cases involving pregnancy or invasive genitourinary procedures, does not necessitate treatment [21]. In the context of complicated UTIs due to underlying anatomical abnormalities, the application of 5–7 days of antibiotic therapy has proven as effective as longer durations (10–14 days) in eight RCTs involving over 1300 patients [22]. When dealing with SSTIs, it is advisable to avoid empiric use of antipseudomonal beta-lactams and/or anti-anaerobic agents unless clinically indicated. Recent guidelines suggest that uncomplicated bacterial cellulitis can be treated effectively with a 5-day course if there is a timely clinical response [21]. Central-venous-catheter-associated bacteremia should be treated for 8 days, acute uncomplicated pyelonephritis for 7 days, and in cases of abdominal infections with prior source control, antibiotics should be administered for 4 days [22]. Additionally, it is crucial to oversee and prevent unnecessary duplication of antibiotics, such as double anti-anaerobic coverage (e.g., piperacillin/tazobactam plus metronidazole). This responsibility can be effectively managed by the hospital pharmacist, a key member of the AS Team.

## 6. Antimicrobial Stewardship in the ICU Setting 

Undoubtedly, patients in the ICU constitute a unique population. They frequently face life-threatening conditions, often present with severe comorbidities, may be immunocompromised, and possess limited physiological reserves. Moreover, their critical illnesses can significantly influence antibiotic pharmacokinetics, resulting in alterations in plasma levels—either increased or decreased, and sometimes both simultaneously. Beyond the patients themselves, the ICU is an exceptional environment due to differences in infections, pathogens, prescribing practices, and the ICU setting itself [61]. Therefore, achieving optimal AS in the ICU extends beyond merely reducing antibiotic use or ensuring adherence to guidelines. It must encompass improved care for individual patients by customizing antibiotic choices and enhancing the overall management of ICU patients as a collective [62]. It is evident that successful AS implementation in the ICU requires a multidisciplinary team. Alongside the ID physician and the clinical pharmacist, a dedicated intensivist in close collaboration with a clinical microbiologist and an infection prevention specialist, as well as a trained nurse specializing in antimicrobial therapy, are indispensable team members [63]. In the recently published EUROBACT-2 study, which reflects all previous statements, epidemiology and outcomes of HA-BSI in ICU patients were investigated [64]. The study was prospective including an international cohort of 2600 patients from 333 ICUs in 52 countries and lasted from June 2019 to February 2021. Among the included patients, 78% had HA-BSI that was ICU-acquired with a SOFA score of 8, whereas 26.7% suffered from pneumonia, and in 26.4%, bacteremia was catheter-related, followed by the abdomen (15%). Authors concluded that prevention of antimicrobial resistance should be based on adequate antimicrobial therapy and source control, optimizing patients’ management and outcomes. It was also evident in the EUROBACT-2 study that optimization of antimicrobial therapy in the critically ill requires a multifaceted approach, among which pharmacokinetic/pharmacodynamic (PK/PD) optimization based on optimal dosing, delivery, and adequate exposure at the source of infection is of paramount importance. On the other hand, the implication of XDR pathogens is an imminent risk for the ICU patient associated with duration of hospital stay, comorbidities, and invasive devise use, whereas antibiotic overuse very often is the result of the hidden fear of the intensivist not to miss an infection.

It is disconcerting to note that up to 70% of ICU patients receive antibiotics, with 30–60% of these prescriptions deemed inappropriate, unnecessary, or suboptimal [65,66]. Consequently, the application of AS in the ICU is particularly warranted, with the primary objectives being to reduce inappropriate antibiotic exposure, mitigate antibiotic resistance, prevent resistance development, improve patient outcomes, reduce side effects, and curtail costs [67]. One notable example comes from an Italian ICU, where the implementation of a PAF system had a profound impact on the collaboration between ID physicians and intensivists regarding antibiotic utilization and patient outcomes [68]. The consumption of broad spectrum β-lactams (*p* = 0.008), carbapenems (*p* = 0.0013), vancomycin (*p* = 0.004), and metronidazole (*p* = 0.0004) was significantly reduced, whereas simultaneously, the days of mechanical ventilation (*p* = 0.0053), days of hospitalization (*p* = 0.0188), and mortality (*p* = 0.0367) were also significantly decreased. Authors pointed out that PAF strengthened the relations as well as the co-operation among the different medical specialties in the ICU. 

In a “Before and After Study” assessing the effectiveness of an AS program in critical care, successful outcomes were attributed to an AS leader who was an intensivist with expertise in Infectious Diseases, particularly trained in managing infections in critically ill patients [69]. The interventions led by this AS leader included daily inspections and interventions conducted twice daily during the morning and afternoon working hours. Furthermore, the implementation of a computerized antibiotic support system has the potential to improve susceptibility to Gram-negative bacteria when feasible [48].

Recognizing that approximately 50 conditions can mimic sepsis [70], the AS Team should always question whether the patient under discussion is truly infected or not [71]. Rapid diagnostics play a pivotal role in the ICU [72], making modern diagnostic microbiology laboratory techniques essential for managing critically ill patients effectively. Overall, the ICU represents one of the most challenging areas within the hospital, necessitating a wide array of AS activities [73]. 

The concept of de-escalation of antibiotics (ADE) has been extensively studied, particularly in ICU patients under the guidance of AS Teams [22,74,75,76,77]. ADE refers to either discontinuing at least one empirically prescribed antimicrobial agent, resulting in a reduced number of administered antibiotics, or switching to an antibiotic with a narrower spectrum of activity within 3–5 days after the initiation of empirical therapy [73]. Internationally, ADE has been recognized as a key component of AS [75,76].

Routsi et al. [77], in the effort to assess the feasibility and the impact of antimicrobial de-escalation in 12 multidisciplinary Greek ICUs with a high burden of antimicrobial resistance, organized from November 2016 to February 2018 a multicenter, prospective, observational study with septic patients and documented infections. A total of 262 Greek patients were included, with a resistant pathogen in 62.9%, of which 49% were characterized as XDR strains. In 97 (37%), de-escalation was judged as not feasible, whereas in the remaining 165, it was applied in 60 (22.9%) within 5 days after the start of empirical therapy. The latter were matched with the application of a propensity score on the day of initiating de-escalation, to an equal number of patients without de-escalation. Bacteremia was encountered in 36.7%, with lung infection in 48.3%, whereas 36.7% were in septic shock when starting empiric therapy. In 21.7%, *A. baumannii* was isolated; in 18.3%, *K. pneumoniae*; in 11.7%, *P. aeruginosa*; *and in 15%*, *E. coli*. A lower all-cause 28-day mortality was observed (13.3% versus 36.7%, *p* = 0.006), whereas de-escalation was characterized as a significant factor for 28-day survival (*p* = 0.005). The study explored why intensivists may be hesitant to engage in de-escalation, with one of the main reasons being a reluctance to change a treatment plan that seems to be working well, along with concerns about the safety of de-escalation [77]. The DIANA study (determinants of antimicrobial use and de-escalation) [78], along with the study by Routsi et al. [77], are the most recent prospective studies on de-escalation. However, the DIANA study takes a different approach compared to the Routsi et al. study [77]. It specifically focuses on empirical therapy initiated within 3 days following the prescription of antibiotics and aims to assess the impact of ADE on a clinical cure by day 7 after initiating therapy. The study lasted from October 2016 to May 2018 and involved 1495 patients from 152 ICUs across 28 countries. Unfortunately, ADE was infrequently applied, with only 16% of critically ill patients undergoing this intervention. While no harmful effects of ADE compared to no ADE were observed, it is important to note that residual confounding factors could not be entirely excluded. It is evident that the field of de-escalation remains open and requires further investigation across various medical sub-specialties, both in the ICU and non-ICU patient populations.

## 7. Applying Antimicrobial Stewardship in Surgery

The CDC has established criteria for defining surgical site infections (SSIs) as “an infection related to an operative procedure that occurs at or near the surgical incision (incisional or organ/space) within 30 days of the procedure or within 90 days if prosthetic material is implanted during surgery” [79]. Organ/space SSIs, while accounting for only one-third of all SSIs, are responsible for more than 90% of SSI-related deaths [80]. SSIs are a common occurrence, contributing to 38% of nosocomial infections, and affecting 2–5% of the over 30 million patients undergoing surgical procedures annually in the United States [81,82,83]. However, it is estimated that approximately 60% of SSIs could be prevented through the consistent application of preventive measures recommended with clinical guidelines and protocols when executed correctly [80,82]. To effectively address the prevention of SSIs, the formation of a Surgical Antibiotic Stewardship Interdisciplinary Team (SAS) within the main hospital antibiotic stewardship team is crucial. The SAS should be composed of the following members: (i) an ID specialist dedicated to AS serving as the leader; (ii) surgeons representing the main surgical departments; (iii) a clinical pharmacist; (iv) the head nurse of the operating room; and (v) an anesthesiologist.

The primary objective of the SAS is to optimize surgical prophylaxis and antimicrobial therapy. The team should actively engage in and oversee the following aspects:

### 7.1. Antimicrobial Prophylaxis

#### 7.1.1. Operative Procedures That Require Prophylaxis

Antimicrobial prophylaxis in surgery is directed towards clean-contaminated wounds, defined as operations involving the entry of a viscus during surgery under controlled conditions and without unusual contamination [84]. Clean-contaminated operations carry an SSI risk ranging from 2.4% to 7.7% [84]. It also includes the implantation of foreign materials in high-risk patients, such as those with extreme age, poor nutritional status, diabetes, obesity, immunosuppression, or fecal colonization with XDR Gram-negative bacteria.

#### 7.1.2. Choice of Antimicrobial Agent

The selection should be based on the narrowest spectrum capable of covering the most likely pathogens associated with specific surgical sites. Broad-spectrum antibiotics do not necessarily result in lower rates of postoperative SSIs compared to narrower-spectrum agents. Therefore, cefazolin, a first-generation cephalosporin, or cefuroxime, a second-generation cephalosporin with a broader spectrum of activity and a similar half-life of approximately 2 h, are typically the preferred antibiotics of choice for surgical prophylaxis [84]. Both of these antibiotics offer an excellent safety profile and demonstrate high activity against the pathogens commonly associated with SSIs, including Enterobacterales and, notably, *E. coli*, as well as skin flora such as *Streptococci*, methicillin-sensitive *S. aureus* (MSSA), and coagulase-negative *Staphylococci* (pathogens that are capable of developing infections whenever foreign bodies are implanted) [84,85,86]. In light of the global increase in fecal carriage of ESBLs and based on evidence indicating that ESBL carriage more than doubles the risk of SSIs (7.2% in carriers versus 1.6% in non-carriers), there has been a suggestion that surgeons should consider conducting preoperative surveillance for fecal colonization with ESBL-producing bacteria and carbapenemases (KPC/MBL). This surveillance is particularly relevant for high-risk patients, such as those undergoing colorectal surgery, prostate biopsy, or liver transplantation [87]. In the event of a positive result, the SAS Team should determine the most appropriate antibiotic for prophylaxis, as ESBLs have the capability to hydrolyze all cephalosporins.

#### 7.1.3. Optimal Dose

Achieving the right drug concentration at the surgical site, including weight-based dosing, is crucial for effective surgical prophylaxis. Vancomycin may be considered as an alternative to cephalosporins in specific circumstances [84,88,89]: (i) if a hospital has experienced a cluster of SSIs caused by MRSA or methicillin-resistant coagulase-negative *Staphylococci;* (ii) when a patient is known to be colonized with MRSA; and (iii) in the case where a patient is at a high risk for MRSA colonization, especially when surveillance data are unavailable, such as recent hospitalization, residing in a nursing home, undergoing hemodialysis, or receiving immunosuppressive medications. In such cases, a beta-lactam antibiotic (either a first- or second-generation cephalosporin) may be added to vancomycin due to its effectiveness against Gram-negative bacilli. However, for patients with allergies to cephalosporins, alternatives such as gentamicin, ciprofloxacin, levofloxacin, or aztreonam can be considered [88,89]. When there is a known colonization or recent infection with an MDR or XDR pathogen, the selection of antibiotics should be personalized based on the specific circumstances [84].

#### 7.1.4. Timing of Antibiotics Administration

Prophylaxis should be initiated within 60 min prior to surgical incision in order for adequate drug tissue levels of the pre-administered antibiotic, to be obtained at the time of the initial incision [84,90,91,92]. Therefore, the half-life of the administrated antibiotic should be considered [82]. To be also pointed out is that for vancomycin or fluoroquinolones, prophylactic administration should begin 120 min before surgical incision because of the 1 h prolonged infusion required for those antibiotics [91].

#### 7.1.5. Duration of Surgical Prophylaxis

Generally, repeating antimicrobial dosing after wound closure is not recommended [93]. Randomized clinical trials have shown no significant difference in the rate of SSIs between single-dose prophylaxis and multiple-dose regimens administered for less than or more than 24 h [80]. However, some evidence suggests that prolonged postoperative administration of antibiotics, not exceeding 24 h postoperatively, may be beneficial in specific surgical procedures. This includes cardiac, vascular, and orthopedic surgeries, including arthroplasty, as well as prosthetic surgeries, compared to single-dose prophylaxis [80]. To ensure adequate antimicrobial serum and tissue concentrations, repeating intraoperative dosing should be considered only for procedures that exceed two half-lives of the administered antibiotic and for operations involving excessive blood loss (more than 1500mL) [84]. Redosing may also be necessary in situations where factors shorten the antimicrobial half-life of the administered antibiotic, such as extensive burns [80,84].

#### 7.1.6. Decolonization of *S. aureus*

The application of nasal mupirocin and chlorhexidine gluconate bathing for a duration of 5 days has been shown to reduce the risk of surgical site infections, particularly in patients undergoing arthroplasty, cardiac surgery, or procedures involving foreign body implantation [82,94]. However, there is no consensus regarding the benefit of preoperative screening for *S. aureus* colonization [82,94].

Unfortunately, errors in the selection, timing, dosing, and duration of surgical prophylaxis are prevalent in healthcare settings. A study involving 34,133 patients undergoing surgery across the United States revealed that only 56% of patients received antimicrobial prophylaxis within 1 h before incision, and antimicrobials were discontinued within 24 h of surgery in just 41% of patients [95]. Furthermore, a systematic review and meta-analysis involving 54,552 patients from 14 research papers found that the risk of SSIs was nearly doubled when SAP was administered after incision and increased five-fold when given more than 120 min prior to incision [96]. It is crucial to emphasize that incisional wound irrigation before closure should not be performed for the purpose of preventing SSIs. This procedure lacks benefits in SSI prevention and is associated with the emergence of antibiotic resistance [93,96]. In Figure 1, a proposal from the “Global Alliance for Infections in Surgery” is presented, outlining Surgical Care Bundles that should be implemented by the Surgical Antibiotic Stewardship Interdisciplinary Team (SAS Team) to prevent SSIs. Based on the previously mentioned information, it is evident that the SAS Team should create a comprehensive booklet on surgical prophylaxis, incorporating specific guidelines tailored to various types of surgeries [97]. This booklet can serve as a valuable resource for healthcare professionals involved in surgical prophylaxis, helping to reduce errors and improve patient outcomes.

### 7.2. Antimicrobial Therapy

When it comes to the optimal treatment of surgical infections, two fundamental aspects are of paramount importance: source control and the judicious selection of antibiotics. These factors should include considerations not only related to antibiotic choice but also correct dosing, mode of administration, and treatment duration. Timely microbial isolation and antimicrobial testing are also imperative. These principles form the backbone of effective management for surgical infections. However, in the context of applying AS to surgical infections, the focus should extend beyond these fundamental principles. It should also comprise the selection of the most appropriate antibiotic based on several crucial criteria, including spectrum of in vitro activity, pharmacokinetics, safety profile, and optimal in vivo efficacy. This choice can involve the selection of either monotherapy or combination therapy, especially when dealing with infections that may require activity against anaerobic bacteria. Furthermore, the aim should involve a concerted effort to reduce existing resistance rates and further development of antibiotic resistance.

To effectively implement SAS in antibiotic selection for surgical infections, it is essential to adopt strategies similar to those applied in various Internal Medicine Departments (e.g., restrictive antibiotic formulary with preauthorization, prospective audit and feedback, and Facility-Specific Guidelines). These strategies include the following principles:I.The appropriate source control by identifying and eliminating the source of infection or reducing the bacterial load particularly in intra-abdominal and soft tissue infections. These techniques include drainage of abscesses or infected fluid collections and debridement of necrotic tissues, applying both operative and non-operative techniques as soon as possible, particularly in the critically ill patients [80].II.The choice of empirical antibiotic therapy, which should be based on local epidemiology, individual patient risk factors for DTR [11], severity of infection, and infection source.III.The necessity of obtaining appropriate culture specimens in the operation room for direct Gram staining and pathogen identification as well as for susceptibility testing, prior to antibiotic initiation [98].IV.The duration of postoperative therapy for intra-abdominal infections, which according to current guidelines and in case of adequate source control, should not exceed 4 days [99].V.The prediction of resistant pathogens while awaiting culture results, e.g., infection acquired in a healthcare setting, recent administration of antibiotics, as well as the underlying immune status of the infected host.VI.The substitution with targeted antimicrobial therapy as soon as possible when culture results and susceptibility testing are available.

## 8. Redefining the Role of Microbiology Lab in the Application of Antibiotics Stewardship in the Hospital Setting: The Diagnostic Stewardship

Effective collaboration between the AS Team and the Microbiology Lab is crucial for the successful implementation of AS initiatives and protocols [22]. The seamless coordination between these two departments is essential for the development, adaptation, and auditing of guidelines. Therefore, fostering a strong working relationship between the AS Team and the Microbiology Lab staff is highly beneficial, allowing for the integration of traditional and advanced laboratory methods to address the evolving challenges posed with Infectious Diseases in the 21st century [100]. Given that prompt initiation of antibiotics is imperative, especially in critically ill septic patients, the AS Team, in partnership with hospital administrators, has a legitimate reason to request expedited processes from the laboratory. Timely microbiology results play a pivotal role in AS programs, as they offer collaborative opportunities to enhance patient outcomes while concurrently reducing antibiotic consumption [101]. The collaboration between the AS Team and the Microbiology Lab is essential for optimizing patient care, reducing antibiotic resistance, and improving outcomes in cases of Infectious Diseases. By employing advanced laboratory methods and provoking effective communication, these two departments can work together to enhance the quality of care for patients.

Several critical methods should be employed to facilitate this collaboration:**Rapid Bacterial Identification.** Implementing PCR techniques alongside conventional cultures and stains can expedite the identification of bacterial isolates in various specimens such as rhinopharyngeal, bronchial, blood, CSF, and fecal samples, with results available in less than an hour [100,101,102].**MALDI-TOF MS Bacterial Species Identification.** This technique allows for precise identification of bacterial species in under 30 min. The impact of MALDI-TOF MS plus stewardship interventions in patients with bacteremia or candidiasis was evaluated by Huang et al. [103]. MALDI-TOF MS results followed by real-time notification to a member of the AS Team, when compared to traditional methodology, improved time to initiate optimal antibiotic treatment (80.9 vs. 23hours; *p* < 0.001), whereas during the intervention period, mortality was lower (21% vs. 8.9%, *p* = 0.01). Therefore, authors recommended that AS programs in combination with rapid diagnostics were beneficial in terms of the patient outcome. A major advantage is the determination of underlying resistance mechanisms at the level of ESBL and carbapenemase products, aiding in the decision to administer advanced antibiotics while awaiting susceptibility test results [100,101].**Rapid Antibiotic Susceptibility Testing.** Rapid diagnostics that enable tailoring of therapy on the same day blood cultures turn positive have been developed. Commercially available options like the Accelerate Pheno^®^ system (Accelerate Diagnostics, Tucson, Arizona, United States) and VITEK^®^ REVEAL™ (Bio-Merieux, Marcy l’Étoile, France, Europe) offer results in approximately 6 h and 4.5 h, respectively [104,105,106]. These tests are particularly beneficial in cases of bloodstream infections and sepsis, where immediate initiation of effective antimicrobial therapy is critical.**The Selective Reporting of Antibiotic Susceptibility Test.** The IDSA recommends the practice of selective and cascade reporting of antibiotics rather than reporting results for all tested antibiotics [107]. This approach encourages reporting of antibiotics that are specifically suitable for the site of infection, or prioritizing “narrower spectrum agents over broad-spectrum ones” [107]. However, the IDSA characterizes these recommendations as weak and based on low-quality evidence, indicating a need for further data. It is important to note that the Microbiology Lab should provide the AS Team with cumulative antimicrobial susceptibility reports for bacterial isolates, ideally on a semi-annual or annual basis, and separately for each hospital clinical department [108,109]. This reporting strategy empowers the AS Team to engage in discussions and share insights with individual-clinical-department medical staff, influencing their antibiotic selection decisions.**Biomarkers.** Procalcitonin (PCT) is a valuable biomarker that becomes elevated during systemic inflammation and therefore can help differentiate between bacterial and viral infections. It plays a role in assessing the likelihood of a bacterial infection’s presence and, notably, guiding the cessation of antibiotic treatment [110]. PCT, however, should not be relied upon as the sole determinant for initiating empirical antibiotic therapy. Elevated PCT levels can also result from conditions such as severe trauma, surgery, burns, cardiac shock, malaria, systemic vasculitis, and end-stage renal disease, while it may remain negative in localized infections or when measured too early. PCT levels of less than 0.3 mg/L, between 0.3 mg/L and 0.5 mg/L, or a drop of 80% or more from the initial abnormal value may encourage the discontinuation of antibiotics. However, if a patient remains clinically unstable, the continuation of antibiotic therapy should be considered. It is evident that the major contribution of PCT values lies in facilitating the discontinuation of antibiotics, a critical task that should be integrated into the responsibilities of the AS Team, particularly in the ICU.**Next-Generation Sequencing.** Next-generation sequencing (NGS) technologies have become increasingly available for use in the clinical microbiology diagnostic environment. There are three main applications of these technologies in the clinical microbiology laboratory: whole genome sequencing (WGS), targeted metagenomics sequencing, and shotgun metagenomics sequencing. These applications are being utilized for initial identification of pathogenic organisms, the detection of antimicrobial resistance mechanisms, and for epidemiologic tracking of organisms within hospital systems [111]. In the context of diagnostic stewardship, NGS technologies can be used to optimize antimicrobial use. For instance, NGS-based rapid diagnostic tests can help identify the resistance genes in bacteria, leading to a more targeted and effective antibiotic. Moreover, NGS can also be used to predict the susceptibility and resistance of certain bacteria to specific drugs [112]. NGS technologies offer significant potential in diagnostic stewardship, particularly in the areas of antimicrobial resistance surveillance and management. By providing detailed information about the genetic makeup of bacteria, NGS can help clinicians make more informed decisions about treatment, ultimately improving patient outcomes [111,112].

Despite the discovery of attractive newer methodologies, it remains important for the AS Team to advocate for the continued use of Gram staining in appropriate specimens. Gram staining, with a turnaround time of less than 3 min, remains indispensable and irreplaceable even in the context of modern techniques. Furthermore, it is crucial for the Microbiology Lab to consistently assess the quality of specimens, which can lead to specimen rejection and the request for a new, appropriate specimen when necessary. This quality control process helps maintain the reliability and accuracy of diagnostic results.

## 9. Redefining the Role of Registered Nurse in Hospital Antibiotic Stewardship Providers

In recognition of the pivotal role nurses play in patient care, hospital quality improvement, and the urgent need to expand AS efforts, the CDC, in collaboration with the American Nurses Association (ANA), convened a Workgroup comprising registered nurses. This initiative aimed to explore the role of nurses in AS and identify key areas where they could contribute significantly [113]. The Workgroup conducted a series of virtual meetings, culminating in a 1-day live seminar held in July 2016 at the ANA headquarters in Silver Spring, MD. Participants were selected by ANA and CDC for their expertise and interest in AS. The overarching goal of this effort was to determine how nurses could become more engaged and assume leadership roles to bolster AS initiatives in the United States. Subsequently, a White Paper was published in 2019 to inform registered nurses in the United States about the critical issue of antibiotic resistance and to delineate their potential roles in AS [113,114,115,116]. The White Paper identified several key stewardship activities in which nurses could play instrumental roles, working in tandem with other AS Team members [114]. The following stewardship activities have been defined as important functions in which nurses can be involved in operation with the AS Team members [114]: appropriate patient triage and isolation, obtaining early radiable and appropriate cultures, checking timely antibiotic initiations, defining accurate antibiotic allergy history, antibiotic adjustment based on microbiology reports, confirming adverse events, checking antibiotic orders, preparing the antibiotic resistance list, including superinfections, evaluation of length of hospital stay, involvement in de-escalation and programs of patient education, medication reconciliation and transition of an iv to po antibiotic, starting antibiotics promptly in case of sepsis, and identification of *C. difficile* infections and their prompt isolation. Moreover, nurses can contribute to educating healthcare staff on proper catheterization techniques for veins and the urinary bladder, inspecting compliance with surgical prophylaxis protocols, and promoting hand hygiene practices. It is worth noting that AS programs demonstrate greater efficacy when implemented alongside infection control (*p* = 0.030) and hand hygiene interventions (*p* < 0.0001) [56]. These responsibilities necessitate specialized education and training in Infectious Diseases and antimicrobial chemotherapy, underlining the indispensable role of nurses within the AS Team [117]. Consequently, many hospitals have already officially designated a nurse as part of their AS Team.

## 10. Other Antimicrobial Stewardship Interventions

### 10.1. Assessing Penicillin Allergy

Approximately 10–15% of hospitalized patients in the United States report being allergic to penicillin [118,119]. However, it is crucial to recognize that less than 1% of the United States population has experienced a severe reaction, such as anaphylaxis, that would genuinely contraindicate treatment with a beta-lactam antibiotic [22]. This discrepancy between reported allergies and actual severe reactions can complicate the selection of the most appropriate antimicrobial therapy [120]. Therefore, it is essential for AS Teams to collaborate with allergy specialists to implement antibiotic allergy testing protocols. This collaboration ensures that patients are not erroneously excluded from receiving the most appropriate antibiotics [120,121].

### 10.2. Reporting

Data regarding the plans and initiatives of AS Teams, which include antibiotic consumption and resistance, should be disseminated beyond just healthcare providers. It is imperative that this information reaches hospital leadership and key stakeholders, including the pharmacy department, medical staff, various committees, and the hospital board. To ensure effective communication and transparency, regular discussions should be scheduled at predefined intervals following the preparation of comprehensive reports on antibiotic tracking. The insights and conclusions drawn from activities like preauthorization and a prospective audit and feedback should be shared directly with prescribers. This direct feedback loop helps healthcare professionals understand the impact of their prescribing decisions and encourages better antibiotic stewardship practices. Additionally, AS Teams can consider alternative methods of disseminating data and conveying important messages to the hospital staff. This may involve utilizing internal communication channels such as staff newsletters and email notifications to ensure that the broader healthcare community is informed and engaged in the ongoing efforts to optimize antibiotic use and combat antibiotic resistance [22].

### 10.3. Education

Education plays a crucial role in AS, especially for smaller hospitals where personalized AS training is essential. AS Teams, consisting of pharmacists and Infectious Diseases physicians, can serve as valuable sources of stewardship knowledge, providing education to nursing and pharmacy staff, as well as relevant students within the healthcare system [22]. Online education offers an alternative approach to reach a broader audience [122,123,124,125]. However, small group educational programs have proven to be more effective than seminars, mailing campaigns, or the dissemination of guidelines [122]. It is equally important to educate patients and their families on proper antibiotic use. Resources such as downloadable brochures from organizations like the CDC, entitled “Get Smart, Know When Antibiotics Work,” and the ECDC, as well as WHO, are valuable tools for patient education. Other useful examples of implementing educational AS strategies for hospital personnel include regular updates of resistance rates through channels like blogs, websites, newsletters, and the internet, as well as one-on-one provider education, such as academic detailing. Mandatory education for new employees, including physicians, pharmacists, and nursing staff, sharing patient stories illustrating the impact of MDR/XDR pathogens and antibiotic side effects (e.g., *C. difficile* infections) and providing examples of how programs like PAF and “Restricted Formulary and Preauthorization” are applied enhance the education of hospital personnel. Creating hospital-specific messages for display on TV screens in lobbies, cafeterias, patients’ rooms, and computer screens in patient wards and regular evaluation of the program’s results at predefined intervals are essential. The AS Team should also apply a program on stewardship metrics in order for the results of the application of the program to be evaluated at regular time intervals [22,126]. It is essential to emphasize that the presence of engaged and dedicated Infectious Diseases physicians in hospitals is critical for the successful implementation of AS Teams. Hospital administrators should share responsibility for supporting and promoting AS efforts within the healthcare system [21,22].

## 11. Conclusions

In light of the significant surge in global antibiotic resistance, it has become abundantly clear that ASPs should be implemented across all hospital settings, regardless of their size or scientific orientation. These ASPs should encompass a comprehensive range of interventions led by a multidisciplinary team, with a dedicated Infectious Diseases physician serving as a leader, and should foster collaboration with hospital administrators. A substantial body of evidence has unequivocally demonstrated the efficacy of ASPs in mitigating antibiotic resistance, particularly against multidrug-resistant Gram-negative bacteria, including carbapenemase and ESBL producers, as well as in limiting the spread of antibiotic-resistant strains among hospitalized patients. Additionally, ASPs have demonstrated their ability to assist physicians in enhancing the quality of care and ensuring patient safety. One particularly promising approach within ASPs is the “prospective audit with communication and feedback” method, which enhances the efficacy, safety, and feasibility of Antimicrobial Stewardship while fostering positive relationships with medical staff. Furthermore, the concurrent implementation of hand hygiene practices has exhibited a synergistic effect and is strongly recommended for future planning. The dearth in support for advanced research in the field of Antimicrobial Stewardship underscores the need for high-quality studies conducted through international collaboration, which would be of paramount importance to public health. Nonetheless, the existing body of evidence and knowledge is more than sufficient to mandate the obligatory implementation of ASPs, making them a top priority and an imperative. The escalating threat of antibiotic resistance poses a grave and rapidly escalating menace to humanity, demanding immediate action.

## Figures and Tables

**Figure 1 antibiotics-12-01557-f001:**
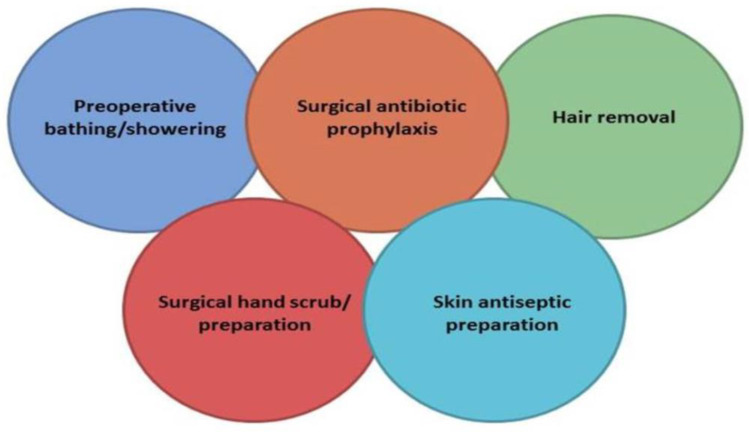
A proposal for a bundle for the prevention of surgical site infections worldwide implemented by Global Alliance for Infections in Surgery. Reproduction after permission [97].

**Table 1 antibiotics-12-01557-t001:** Risk factors for colonization and subsequent infection from Carbapenem-Resistant Enterobacterales [13,21,22].

Administration of antibiotics (mostly carbapenems and quinolones) as well as combinations of multiple antibiotic regimens the preceding 90 days Hospitalization in an ICU ≥ 2 days in the preceding 90 days Known colonization with an XDR strain or preceded infection with a Gram-negative strain producing carbapenemase Hospitalization in the same ward with a patient who is a carrier or is infected from a bacterium producing carbapenemase Resident of a Rehabilitation Center or Home for the Elderly Septic shock Immunosuppression due to hematologic malignancies, neutropenia, bone marrow and solid organ transplantation, solid tumor under chemotherapy, administration of immunosuppressive drugs, chronic administration of steroids (>10 mg of prednisone daily or cumulative total dose of >700 mg for at least 3–4 weeks)

**Table 2 antibiotics-12-01557-t002:** Restrictive Antibiotic Formulary Applied in most Tertiary Hospitals in Greece.

Protected Antibiotics
Carbapenems (Meropenem, Imipenem, Ertapenem) Tigecycline Colistin Fosfomycin Ceftazidime/Avibactam Meropenem/Vaborbactam Imipenem/Citastatin/Relebactam

**Table 3 antibiotics-12-01557-t003:** Successful restrictive policy leads to a reduction in consumption and endurance at the Sismanoglio. Reproduction after permission [57].

	*Pseudomonas aeruginosa*, resistance rates (%)		*Klebsiella Pneumoniae*, resistance rates (%)	
	Before	After	Before	After
Number of Strains	694	372	213	99
Gentamicin	52	37 ^a^	24	13 ^a^
Amikacin	49	31 ^a^	17	11
Ciprofloxacin	55	35 ^a^	17	16
Ceftriaxone	NA	NA	29	15 ^a^
Cefepime	56	31 ^a^	37	12 ^a^
Ceftazidime	42	24 ^a^	31	15 ^a^
Piperacillin/Tazobactam	50	30 ^a^	34	20 ^a^
Aztreonam	62	39 ^a^	29	16 ^a^
Imipenem	10	3 ^a^	0	0

NA, not applicable. ^a^
*p* < 0.05.

**Table 4 antibiotics-12-01557-t004:** Resistance rates (%) during the study period for *Pseudomonas aeruginosa* isolates at Zanion Hospital in Greece. Reproduction after permission [58].

Antibiotic	Year					
	2015	2016	2017	2018	2019	*p* value
	n = 210	n = 219	n = 322	n = 312	n = 309	
Amikacin	45.9	51.2	34.4	33.2	15.6	<0.0001
Ciprofloxacin	53.3	59.1	41.3	51	24.9	0.0003
Cefotaxime	55.0	61.2	23.4	28.9	13.2	<0.0001
Cefepime	49.2	54.0	30.7	28.8	13.4	<0.0001
Piperacillin/Tazobactam	52.1	48.9	23.3	20.3	9.8	<0.0001
Meropenem	57.3	63.1	37.2	42.7	22.6	<0.0001
Imipenem	61.1	71.2	38.8	47.2	25.2	<0.0001

**Table 5 antibiotics-12-01557-t005:** Resistance Rates (%) during the study period for *Klebsiella pneumoniae* isolates at Zanion Hospital in Greece. Reproduction after permission [58].

Antibiotic	Year					
	2015	2016	2017	2018	2019	*p* value
	n = 467	n = 346	n = 431	n = 512	n = 622	
Amikacin	55.4	31.5	31.3	44.9	27.2	<0.0001
Ciprofloxacin	81.1	82.3	73.8	68.1	70.1	0.518
Cefotaxime	81.8	84.0	71.6	67.9	67.7	0.0665
Cefepime	83.5	75.5	68.0	64.0	62.7	0.0061
Piperacillin/Tazobactam	81.8	86.2	69.3	62.1	65.8	0.0332
Meropenem	81.2	85.6	66.3	59.7	61.8	0.0087
Imipenem	76.5	81.8	65.1	59.6	62.2	0.0519
Tigecycline	26.4	16.2	3.9	8.3	19.2	0.1855

## Data Availability

Data sharing not applicable. The reproduction of Table 3 is authorized under License Number 5626000802406, while Table 4 and Table 5 are permitted for reproduction with License Numbers 5624660098009. These permissions were obtained through RightsLink. Figure 1 was reproduced with permission from Dr. Massimo Sartelli and can be accessed via the following link: https://infectionsinsurgery.org/global-alliance-for-infections-in-surgery-bundles-for-the-prevention-of-surgical-site-infections-worldwide/.

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
