# Peer review of "Antimicrobial Stewardship in the Hospital Setting: A Narrative Review"

_antibiotics, 2023, doi:10.3390/antibiotics12101557_

Round 1

Reviewer 1 Report

The authors have addressed an important health issue. There are few concerns over the review;

1. The methodology section is missing. The methodology section should describe how the data was collected. What was source of data collection (PubMed etc.,). What was the time period and range for collecting data(e.g. related articles published in between 2000 to 2023 were studied). The inclusion or exclusion of criteria of publication for review. 

2. The tables in the articles are reproduced from previously published articles (this may be considered as plagiarism). The authors must design their own tables from the already published data. Permission is required from publisher. e.g. table 3, in current review is authors own data published in "Scandinavian Journal of Infectious Diseases", however reproducing it, they need permission of the publisher.

The organization of the review is haphazard. 

If the authors do not change the tables in paper, the review should rejected. 

language is ok

Author Response

  1. The methodology section is missing. The methodology section should describe how the data was collected. What was source of data collection (PubMed etc.,). What was the time period and range for collecting data(e.g. related articles published in between 2000 to 2023 were studied). The inclusion or exclusion of criteria of publication for review.

Answer: Thank you for your comment. A description of the data collection was added to accommodate to reviewers comment.

  1. The tables in the articles are reproduced from previously published articles (this may be considered as plagiarism). The authors must design their own tables from the already published data. Permission is required from publisher. e.g. table 3, in current review is authors own data published in "Scandinavian Journal of Infectious Diseases", however reproducing it, they need permission of the publisher.

Answer: Thank you for your comment. We have obtained the necessary permissions for the reproduction of all tables, including Figure 1. To provide further details, License Number 5626000802406 has been secured for Table 3, while License Numbers 5624660098009 have been acquired for Tables 4 and 5. These permissions were granted through the RightsLink Printable License and have been successfully uploaded along with the submission of the manuscript. 

The organization of the review is haphazard.

Answer: We made efforts to restructure the manuscript for improved readability.

If the authors do not change the tables in paper, the review should rejected.

Answer: We have obtained the necessary permissions for the reproduction of all tables, including Figure 1. To provide further details, License Number 5626000802406 has been secured for Table 3, while License Numbers 5624660098009 have been acquired for Tables 4 and 5. These permissions were granted through the RightsLink Printable License and have been successfully uploaded along with the submission of the manuscript.

Reviewer 2 Report

1. The title of the manuscript is somewhat misleading as the body of the manuscript refers mostly to initiatives which could be effective in 
Greece (or a specific hospital) and uses the outcomes from other centers throughout the world to substantiate this. Perhaps, retitling this manuscript to something along the lines of "Antimicrobial Stewardship Initiatives - Lessons from Hospitals in Greece" might be more appropriate. To this end, the authors should indicate early on if this is the case as it is not until page 5 with reference to Table 1 that this is suggested.   

2. While not overtly stated, this manuscript should be highlighted as a narrative review. Even including that statement, some type of methods should be included. At present it appears to be a conglomeration of established policies or initiatives with corroborating data points pertaining to stewardship. 

3. Strongly consider targeting certain areas and discussing while omitting others. At present, the manuscript is too lengthy. 

4. To the points above, I struggled to see how the manuscript added to the current literature regarding stewardship. Perhaps capturing certain areas and sharing the center's experience will help to mitigate this.  

5. The (abstract) conclusion of more stringent policies being necessary is interesting as  one could argue the policies in place which are less stringent often receive "pushback" or are not always effective in many institutions thus making these more difficult might not necessarily translate to better outcomes. 

6. On page 6, lines 227-233, consider referencing the IDSA Gram Negative Resistance Guidelines. See: Tamma PD, Aitken SL, Bonomo RA, Mathers AJ, van Duin D, Clancy CJ. Infectious Diseases Society of America 2023 Guidance on the Treatment of Antimicrobial Resistant Gram-Negative Infections. Clin Infect Dis. 2023 Jul 18:ciad428. doi: 10.1093/cid/ciad428.

These comments on page 6 also feel like an overgeneralization. 

Minor edits are necessary. 

Author Response

Detailed response to reviewer 2

  1. The title of the manuscript is somewhat misleading as the body of the manuscript refers mostly to initiatives which could be effective in Greece (or a specific hospital) and uses the outcomes from other centers throughout the world to substantiate this. Perhaps, retitling this manuscript to something along the lines of "Antimicrobial Stewardship Initiatives - Lessons from Hospitals in Greece" might be more appropriate. To this end, the authors should indicate early on if this is the case as it is not until page 5 with reference to Table 1 that this is suggested.

Answer: Thank you for your valuable comment. Our manuscript represents a comprehensive narrative review on antimicrobial stewardship, introducing innovative methods and addressing the required personnel for implementing these innovations. While we acknowledge the emphasis on examples from Greece within the manuscript, we firmly believe that the principles discussed extend beyond any specific country, including Greece. Consequently, we have made modifications to the manuscript's title, aiming to maintain its broader scope without incorporating a specific country reference.

  1. While not overtly stated, this manuscript should be highlighted as a narrative review. Even including that statement, some type of methods should be included. At present it appears to be a conglomeration of established policies or initiatives with corroborating data points pertaining to stewardship.

Answer: We sincerely appreciate your comment and value your recommendation. In response to the reviewer's feedback, we have included a description of the data collection process in the manuscript.

  1. Strongly consider targeting certain areas and discussing while omitting others. At present, the manuscript is too lengthy.

Answer: We made efforts to restructure the manuscript for improved readability.

  1. To the points above, I struggled to see how the manuscript added to the current literature regarding stewardship. Perhaps capturing certain areas and sharing the center's experience will help to mitigate this.

Answer: In the context of narrative reviews, the primary objective is to systematically compile and present all pertinent available data in a comprehensive manner. Unlike original research articles, the emphasis here lies in meticulously reviewing and synthesizing the existing and current body of knowledge.

  1. The (abstract) conclusion of more stringent policies being necessary is interesting as one could argue the policies in place which are less stringent often receive "pushback" or are not always effective in many institutions thus making these more difficult might not necessarily translate to better outcomes.

Answer: The abstract's conclusion has been revised to avoid suggesting that stringent policies, despite their potential challenges, always result in improved outcomes.

  1. On page 6, lines 227-233, consider referencing the IDSA Gram Negative Resistance Guidelines. See: Tamma PD, Aitken SL, Bonomo RA, Mathers AJ, van Duin D, Clancy CJ. Infectious Diseases Society of America 2023 Guidance on the Treatment of Antimicrobial Resistant Gram-Negative Infections. Clin Infect Dis. 2023 Jul 18:ciad428. doi: 10.1093/cid/ciad428.

These comments on page 6 also feel like an overgeneralization.

Answer: This section was deleted to shorten the manuscript

Reviewer 3 Report

Line 44: Fleming, with his foresight, appeared as a pessimistic prophet to the medical community, however, was also emerged as an optimistic advocate for humanity – please rewrite this sentence for clarity

Line 62: Can authors include the antibiotic categories classes here

Line 190: Initiating an AS program, the first action should be the calculation of the specific needs of the involved personnel. Authors need to provide more details on this otherwise it is vague without examples of such needs.

Line 197: was suggested that only 1.2 and 0.4 FTE pharmacist and physician respectively would be 197 required. For which hospital size?

Line 300: PAF was associated with greater reduction of antibiotics, whereas in another transition… this should be antibiotics usage or prescription or resistance?

Line 684: When it comes to the optimal treatment of surgical infections, two fundamental aspects are of paramount importance: source control and the judicious selection of antibiotics. These factors should include considerations not only related to antibiotic choice but also correct dosing, mode of administration, and treatment duration. Authors should include timely microbial isolation and antimicrobial testing as imperative.

Line 737: Several critical methods should be employed to facilitate this collaboration. Authors need to include a recommendation on the routine use of Next-generation sequencing for surveillance of antimicrobial resistance, pathogenicity and virulence in large hospital microbiology.

Author Response

Detailed response to reviewer 3

Line 44: Fleming, with his foresight, appeared as a pessimistic prophet to the medical community, however, was also emerged as an optimistic advocate for humanity – please rewrite this sentence for clarity.

Answer: The sentence was rephrased to align with the reviewer's suggestion.

Line 62: Can authors include the antibiotic categories classes here.

Answer:  The antibiotic categories/classes have been included in the document as requested.

Line 190: Initiating an AS program, the first action should be the calculation of the specific needs of the involved personnel. Authors need to provide more details on this otherwise it is vague without examples of such needs.

Answer: Thank you for your valuable comment. In addressing the reviewer's request for a more comprehensive elaboration on the calculation of specific personnel needs during the initiation of an Antimicrobial Stewardship program, we have developed the reviewer's feedback. Our response involves a more in-depth explanation of each of these aspects, including specific examples and detailed insights that underscore the importance of these key points.

Line 197: was suggested that only 1.2 and 0.4 FTE pharmacist and physician respectively would be 197 required. For which hospital size?

Answer: The size of the hospital was added to the manuscript.

Line 300: PAF was associated with greater reduction of antibiotics, whereas in another transition… this should be antibiotics usage or prescription or resistance?

Answer: The sentence has been revised, and the data in question pertains to antibiotic usage. The transaction was linked to an increase in the utilization of broad-spectrum antibiotic prescriptions.

Line 684: When it comes to the optimal treatment of surgical infections, two fundamental aspects are of paramount importance: source control and the judicious selection of antibiotics. These factors should include considerations not only related to antibiotic choice but also correct dosing, mode of administration, and treatment duration. Authors should include timely microbial isolation and antimicrobial testing as imperative.

Answer: A modification was made to the sentence in response to the reviewers' suggestion.

Line 737: Several critical methods should be employed to facilitate this collaboration. Authors need to include a recommendation on the routine use of Next-generation sequencing for surveillance of antimicrobial resistance, pathogenicity and virulence in large hospital microbiology.

Answer: A paragraph on diagnostic stewardship has been incorporated, following the reviewer's suggestion, which includes a recommendation for the routine application of NGS in the extensive microbiology surveillance within large hospital settings. This addition aligns with the reviewer's input and enhances the content accordingly.

Round 2

Reviewer 1 Report

The manuscript has been improved

Minor changes required

Author Response

The manuscript has been improved

Answer: thank you for the remarkable comment.

Reviewer 2 Report

1. The authors have added some methods as to how manuscripts were selected, but should also consider exclusion criteria. 

2. Is it possible to quantify how many manuscripts were identified using the search strategy.  

3. In Table 1, clarify total prednisone dose of >700mg. Is this monthly? 

4. In Table 2, carbapenems are listed as a class and then again individually. Please revise. 

Minor editing is required. 

Author Response

  1. The authors have added some methods as to how manuscripts were selected, but should also consider exclusion criteria.

Answer: Exclusion criteria were incorporated in response to reviewer suggestions to enhance the quality of our study.

  1. Is it possible to quantify how many manuscripts were identified using the search strategy.

Answer: We quantified the number of manuscripts identified through our search strategy and determined the total number of manuscripts used in our narrative review.

  1. In Table 1, clarify total prednisone dose of >700mg. Is this monthly? 

Answer: The dose is referred to accumulative dose of > 700mg for 3-4 weeks.

  1. In Table 2, carbapenems are listed as a class and then again individually. Please revise. 

Answer: We made modifications to the table to address the recommendations provided by the reviewer.